# Gonadal Development and Differentiation of Hybrid F_1_ Line of *Ctenopharyngodon idella* (♀) × *Squaliobarbus curriculus* (♂)

**DOI:** 10.3390/ijms251910566

**Published:** 2024-09-30

**Authors:** Qiaolin Liu, Shitao Hu, Xiangbei Tang, Chong Wang, Le Yang, Tiaoyi Xiao, Baohong Xu

**Affiliations:** 1Fisheries College, Hunan Agricultural University, Changsha 410128, China; qiaolinliu2017@hunau.edu.cn (Q.L.);; 2Yuelushan Laboratory, Changsha 410128, China

**Keywords:** hybrid F_1_ offspring of *Ctenopharyngodon idella* and *Squaliobarbus curriculus*, reproductive fertility, transcriptome sequencing, gonadal development, gene expression

## Abstract

The hybrid F_1_ offspring of *Ctenopharyngodon idella* (♂) and *Squaliobarbus curriculus* (♀) exhibit heterosis in disease resistance and also show abnormal sex differentiation. To understand the mechanism behind gonadal differentiation in the hybrid F_1_, we analyzed the transcriptomes of *C. idella*, *S. curriculus*, and the hybrid F_1_; screened for genes related to gonad development in these samples; and measured their expression levels. Our results revealed that compared to either *C. idella* or *S. curriculus*, the gene expressions in most sub-pathways of the SNARE interactions in the vesicular transport pathway in the hypothalamus, pituitary, and gonadal tissues of their hybrid F_1_ offspring were significantly up-regulated. Furthermore, insufficient transcription of genes involved in oocyte meiosis may be the main reason for the insufficient reproductive ability of the hybrid F_1_ offspring. Through transcriptome screening, we identified key molecules involved in gonad development, including HSD3B7, HSD17B1, HSD17B3, HSD20B2, CYP17A2, CYP1B1, CYP2AA12, UGT2A1, UGT1A1, and FSHR, which showed significant differences in expression levels in the hypothalamus, pituitary, and gonads of these fish. Notably, the expression levels of UGT1A1 in the gonads of the hybrid F_1_ were significantly higher than those in *C. idella* and *S. curriculus*. These results provide a scientific basis for further research on the gonadal differentiation mechanism of hybrid F_1_ offspring.

## 1. Introduction

China is known for having some of the most abundant fish resources in the world, making it the leading aquaculture country globally. According to Yue et al. [1], there are over 800 species, and 240 improved varieties have been used in Chinese aquaculture. However, recently, the water quality in China has deteriorated, resulting in an increase in diseases among the main aquaculture species. As a result, breeding improved fish varieties has become a crucial task in promoting the sustainable and healthy development of fisheries. This includes the breeding of high-yield, high-quality, disease-resistant, and stress-resistant varieties, as emphasized by Du et al. [2] and Hu et al. [3].

Hybrid breeding is a widely used method in animal and plant breeding, known for its effectiveness in improving growth rates, meat quality, and disease resistance in fish [4,5]. Through artificial hybridization technology, fish can produce offspring with significant advantages [2]. For instance, a study on a hybrid of *Megalobrama amblycephala* (♀) × *Erythroculter mongolicus* (♂) showed that the average weight of the offspring was significantly higher than that of either parent species [6]. Similarly, a study on the crossbreeding of *Platichthys bicoloratus* (♂) × *Paralichthys olivaceus* (♀) found that the body height of the offspring increased by more than 4% compared to the parents, demonstrating clear growth advantages [7].

Reproductive fertility is one of the advantages of hybrid fish, although the fertility of hybrid offspring varies. For instance, while the hybrid offspring of *Erythroculter ilishaeformis* × *Carassius auratus* [8] and *C. auratus* × *M. amblycephala* [9] were fertile, the hybrid offspring of *Hypophthalmichthys nobilis* × *Squaliobarbus curriculus* [10] were found to be sterile. The mechanisms of sex determination and differentiation in fish are complex and can be influenced by both genetics and the environment [11,12]. Sex-determining genes are activated during the embryonic period and undergo a series of developmental processes to form either a sperm nest or an ovary [13,14]. Sex differentiation and gonadal development in fish are also closely linked to changes in the levels of sex steroid hormones [15]. These hormones are synthesized by steroid synthetase, which includes the cytochrome P450 gene family (CYP), the hydroxysteroid deoxygenase family (HSD), and other steroid oxidoreductases. Initially, StAR transports cholesterol from the cytosol to the mitochondria, where it is then converted to testosterone by steroid synthetase. This testosterone can then be further converted to either 17β-estradiol (E2) or 11-testosterone (11-KT), ultimately resulting in the production of estrogen and androgen [16].

Grass carp (*Ctenopharyngodon idella*) is the main species used in freshwater aquaculture in China. To improve the disease resistance and survival rate of *C. idella*, Jin et al. [17] utilized distant hybridization technology to hybridize *C. idella* and *S. curriculus*. They discovered that the hybrid offspring exhibited similar quantitative traits to their female parent. He et al. [18] also reported that the hybrid F_1_ showed a combination of fast growth inherited from female *C. idella* and strong resistance to GCRV inherited from male *S. curriculus*. Transcriptome sequencing of the gonadal tissues of 12-month-old *C. idella* revealed that *dmrt1* and *Amh* were highly expressed genes in the *C. idella* sperm nest, whereas *CYP19A1A* and *foxl2* were highly expressed in the *C. idella* ovary, playing a role in the early gonadal development of *C. idella* [19]. Our preliminary study showed that the gonadal differentiation of the hybrid F_1_ offspring of *C. idella* (♀) × *S. curriculus* (♂) varied, with some females being partially fertile and others being infertile. The fertility performance of individuals also varied greatly, with diverse degrees of ovarian differentiation. Males were able to produce sperm cells, whereas no mature sperm or normal ejaculation were observed [20]. Anatomical, histological, and production results showed that whereas some females of the hybrid F_1_ line of *C. idella* (♀) × *S. curriculus* (♂) were partially fertile, others were infertile. Similarly, among fertile individuals, some were partially infertile. Males were able to produce sperm cells, but no mature sperm or normal sperm excretion were observed. To uncover the molecular regulatory mechanism of gonadal development in the hybrid F_1_ line of *C. idella* (♀) × *S. curriculus* (♂), transcriptome sequencing was used to screen and detect the expression levels of genes related to *C. idella*, *S. curriculus*, and their hybrid F_1_ offspring. Our results provide valuable data for understanding the gonadal differentiation characteristics of *C. idella*, *S. curriculus*, and their hybrid F_1_ offspring, as well as for the future breeding of parental and hybrid populations.

## 2. Results

### 2.1. Transcript Expression Analysis

The transcriptome sequences and corresponding amino acid sequences were predicted using TransDecoder, and a total of 159,917 open reading frames (ORFs) were obtained, of which 142,391 were complete. A total of 8636 transcription factors were predicted using animalTFDB 2.0 (Figure 1A). In total, 43,776, 123,331, 93,639, and 110,049 isoforms were obtained based on the COG, GO, KEGG, and KOG databases, respectively (Appendix A).

The homologous species were identified through sequence alignment using the NR database. The main homologous species were *Danio rerio*, *Astyanax mexicanus*, *Ctenopharyngodon idella*, *Oncorhynchus mykiss*, *Cyprinus carpio*, *Esox luclus*, *Stegastes partitus*, *Lepisosteus oculatus*, *Oreochromis niloticus*, and *Carassius auratus* (Figure 1B). These species are all fish, indicating the reliability of the sequencing data and the absence of contamination from other species. The decision to compare multiple fish species may have been influenced by the differences in data and parameter settings among the species in the database.

The results of the COG function classification showed that the two highest transcriptional groups were general function prediction (18.26%) and signal transduction mechanisms (14.87%) (Figure 1C). Additionally, the results of the eggnog functional classification indicated that 45.3% of the transcripts were classified as having an unknown function (Figure 1D). This suggested that the eggnog functional classification may not be suitable for classifying the transcriptomes of the studied fish.

### 2.2. Transcriptional Differences between C. idella, S. curriculus, and Their Hybrid F_1_ Offspring

Cluster analysis of the transcriptome data revealed that the transcriptomes of *S. curriculus* were more similar to those of their hybrid F_1_ offspring in the hypothalamus and pituitary tissue compared to *C. idella* (Figure 2A). Additionally, the hypothalamus transcriptomes of *C. idella* showed greater similarity to those of the hybrid F_1_ offspring compared to other samples (Figure 2A). Similarly, the pituitary and gonad transcriptomes of *C. idella* were also more similar to each other than to other samples (Figure 2A). When comparing the transcriptome of *S. curriculus* to that of *C. idella*, the numbers of up- and down-regulated genes were relatively balanced (37,293 vs. 37,197 in the hypothalamus, 41,293 vs. 38,881 in pituitary tissue, and 40,083 vs. 39,446 in gonadal tissue; Figure 2B,E,H). However, the number of up-regulated genes in the transcriptome of the hybrid F_1_ offspring was significantly higher than that in *C. idella* and *S. curriculus* (Figure 2C,D,F,G,I,J). The GO classification results indicated significant differences in the transcription levels of genes involved in cellular components, molecular functions, and biological processes. Specifically, for the cellular component, the transcription levels of genes involved in cells, membranes, macromolecular complexes, organelles, organelle parts, membrane parts, and cell parts accounted for more than 10%. In terms of molecular function, the transcription levels of genes involved in catalytic activity and binding accounted for more than 10%. For biological processes, the transcription levels of genes involved in metabolic processes, cellular processes, signaling, multicellular organismal processes, developmental processes, single-organism processes, response to stimuli, localization, biological regulation, and cellular component organization of biogenesis accounted for more than 10% (Figure 1D).

### 2.3. Screening and Mining of Gonadal-Development-Related Pathways

The pathways associated with reproduction that were screened via transcriptome analysis mainly include the oxytocin signaling pathway (ko04921), the GnRH signaling pathway, SNARE interactions in vesicular transport (ko04130), renin secretion (ko04924), and oocyte meiosis (ko04114) (Table 1).

Compared to *C. idella*, only the gene expression in the VAMP7 sub-pathway of the SNARE interactions in the vesicular transport pathway in the hypothalamus, pituitary, and gonadal tissues of *S. curriculus* was significantly reduced. All other sub-pathways showed a mix of up- and down-regulated genes (Appendix A). However, when compared to either *C. idella* or *S. curriculus*, the gene expressions in most sub-pathways of the SNARE interactions in the vesicular transport pathway in the hypothalamus, pituitary, and gonadal tissues of their hybrid F_1_ offspring were significantly up-regulated. The remaining significantly different sub-pathways also showed a mix of up- and down-regulated genes, with no sub-pathways containing significantly down-regulated genes (Figure 3).

In the oxytocin signaling pathway, the number of up-regulated genes in *S. curriculus* was lower compared to that in *C. idella* in the hypothalamus, pituitary, and gonadal tissues. Additionally, the number of sub-pathways with differentially expressed genes was also lower (Appendix A). In comparison to *S. curriculus*, the expressions of the EEF2 gene in the hypothalamus and pituitary were significantly reduced, whereas the other differential genes showed significant up-regulation or both up- and down-regulation (Figure 4A,B). However, the distribution of these significantly differential genes in sub-pathways across different organs was not consistent (Figure 4A–C). Furthermore, in the hybrid F_1_ offspring, the significantly different genes were either significantly up-regulated or showed both up- and down-regulation simultaneously (action genes) (Figure 4D–F).

In the renin secretion pathway, the expressions of only a few genes in the three organs of *S. curriculus* were significantly different compared to those for *C. idella* (Appendix A). However, in the hybrid F_1_ offspring, the expressions of different genes were significantly up-regulated or both up- and down-regulated (such as *CaM* in the hypothalamus and pituitary gland and *Cn* in the gonadal organs) compared to both *S. curriculus* and *C. idella*. The only exception was the *BKCa* gene, which showed significant down-regulation in the gonadal organ of *S. curriculus* (Figure 5).

In the oocyte meiosis pathway, there were significant differences in the expression of genes between *C. idella* and *S. curriculus* in the hypothalamus, pituitary, and gonadal tissues. Whereas most of the differentially expressed genes in *S. curriculus* were both up- and down-regulated simultaneously, only a few genes showed significant up- or down-regulation (Appendix A). Interestingly, in the hybrid F_1_ offspring, the expressions of differentially expressed genes in the hypothalamus and pituitary were either up-regulated or both up- and down-regulated simultaneously (Figure 6A,B), whereas a large number of genes were significantly down-regulated in the gonadal tissue (Figure 6C). Although the number of down-regulated genes in the gonadal tissue of the hybrid F_1_ offspring was lower than that in *C. idella*, there was a similar trend to that seen in *S. curriculus* (Figure 6D–F). These results suggest that insufficient transcription of genes involved in oocyte meiosis may be the main factor contributing to the reduced reproductive ability of the hybrid F_1_ offspring.

In the GnRH signaling pathway, there were significant differences in the expression of genes between *C. idella* and *S. curriculus* in the hypothalamus, pituitary, and gonadal tissues. These differences were observed in both the up- and down-regulation of genes simultaneously (Appendix A). Furthermore, compared with *S. curriculus* or *C. idella*, the expressions of differentially expressed genes in the hybrid F_1_ offspring’s hypothalamus, pituitary, and gonadal tissues were up-regulated and up- and down-regulated simultaneously (Figure 7A–F).

### 2.4. Expression of Gonadal-Development-Related Genes in C. idella, S. curriculus, and Hybrid F_1_ Offspring

The expressions of the *HSD3B7* gene did not show significant differences in the hypothalamus, pituitary, and gonads between *C. idella* and the hybrid F_1_ offspring (*p* > 0.05). However, in *S. curriculus*, the expression of the *HSD3B7* gene was significantly higher in the pituitary gland compared to the gonads, and the expression in the gonads was significantly higher than that in the hypothalamus (Figure 8A). The expression of the *HSD3B7* gene in the hypothalamus of *C. idella* did not significantly differ from that of *S. curriculus* and the hybrid F_1_ offspring (*p* > 0.05; Figure 8A). In the pituitary gland, the expression of the *HSD3B7* gene in *S. curriculus* was significantly higher than that in *C. idella* and the hybrid F_1_ offspring (*p* < 0.05; Figure 8A), whereas the expression in *C. idella* did not significantly differ from that in the hybrid F_1_ offspring (*p* > 0.05; Figure 8A). In the gonads, the expression of the *HSD3B7* gene in *S. curriculus* was significantly higher than that in *C. idella* and the hybrid F_1_ offspring (*p* < 0.05; Figure 8A), whereas the expression in *C. idella* did not significantly differ from that in the hybrid F_1_ offspring (*p* > 0.05; Figure 8A).

The expression of *HSD17B1* in the pituitary gland of *C. idella* was significantly higher than that in the hypothalamus and gonads (*p* < 0.05; Figure 8B). However, there was no significant difference in expression between the hypothalamus and gonads. In *S. curriculus*, the expression of the *HSD17B1* gene in the gonads was the highest, significantly higher than that in the hypothalamus and pituitary gland. The expression of the *HSD17B1* gene in the hypothalamus did not significantly differ from that in the pituitary gland. In the hybrid F_1_ offspring, there was no significant difference in expression between the hypothalamus, pituitary, and gonadal tissues (*p* > 0.05; Figure 8B). Additionally, there was no significant difference in expression of the *HSD17B1* gene between *C. idella*, *S. curriculus*, and the hybrid F_1_ offspring in the hypothalamus (*p* > 0.05; Figure 8B). In the pituitary gland, there was no significant difference in expression between *S. curriculus* and the hybrid F_1_ offspring (*p* > 0.05; Figure 8B), whereas the expression for both was significantly lower than that in *C. idella* (*p* < 0.05; Figure 8B). In the gonads, there was no significant difference in expression between *C. idella* and the in the hybrid F_1_ offspring (*p* > 0.05; Figure 8B), whereas the expression of the *HSD17B1* gene in *S. curriculus* was significantly higher than that in the hybrid F_1_ offspring (*p* < 0.05; Figure 8B).

The expression of the *HSD17B3* gene in the pituitary of *C. idella* was significantly higher than that in the hypothalamus and gonads (*p* < 0.05; Figure 8C). Additionally, the relative expression level in the hypothalamus was significantly higher than that in the gonads (*p* < 0.05; Figure 8C). Similarly, in *S. curriculus*, the expression of the *HSD17B3* gene in the pituitary gland was significantly higher than in the hypothalamus and gonads (*p* < 0.05; Figure 8C), with the expression in the hypothalamus also being significantly higher than in the gonads (*p* < 0.05; Figure 8C). In the hybrid F_1_ offspring, the expression of the *HSD17B3* gene in the hypothalamus was significantly higher than that in the gonads (*p* < 0.05; Figure 8C), whereas the expression in the pituitary gland did not differ significantly from that in the hypothalamus and gonads (*p* > 0.05; Figure 8B). Furthermore, there were no significant differences in the expression of the *HSD17B3* gene between *C. idella*, *S. curriculus*, and the hybrid F_1_ offspring in the hypothalamus and gonads (*p* > 0.05; Figure 8B). However, in the pituitary gland, the expression of the *HSD17B3* gene was the highest in *C. idellus*, followed by that in *S. curriculus*, and lowest in the hybrid F_1_ offspring, with all three having significant differences (*p* < 0.05; Figure 8C).

The expression of the *HSD20B2* gene was the highest in the pituitary gland of *C. idella*, followed by the hypothalamus, and the lowest expression was found in the gonads. All three tissues showed significant differences (*p* < 0.05; Figure 8D). In contrast, there was no significant difference in the expressions of the *HSD20B2* gene in the three tissues of *S. curriculus*. Additionally, the expression of the *HSD20B2* gene in the hypothalamus of the hybrid F_1_ offspring was not significantly different from that in the pituitary gland (*p* > 0.05; Figure 8D). However, the expression of this gene in the pituitary gland was significantly higher than that in the gonads (*p* < 0.05; Figure 8D). Furthermore, the expression of the *HSD20B2* gene in the hypothalamus of *C. idella* was significantly higher than that in *S. curriculus* and the hybrid F_1_ offspring, and the expression of this gene in the hypothalamus of the hybrid F_1_ offspring was significantly higher than that in *S. curriculus* (*p* < 0.05; Figure 8D). Furthermore, the expression of the *HSD20B2* gene in the hypothalamus of *C. idella* was significantly higher than that in *S. curriculus* and the hybrid F_1_ offspring, and the expression of this gene in the hypothalamus of the hybrid F_1_ offspring was significantly higher than that in *S. curriculus* (*p* < 0.05; Figure 8D). Interestingly, there was no significant difference in the expression of the *HSD20B2* gene in the gonads between *C. idella*, *S. curriculus*, and the hybrid F_1_ offspring (*p* > 0.05; Figure 8D).

The expression of *CYP17A2* gene in the hypothalamus of *C. idella* was significantly higher than that in the pituitary gland and gonads (*p* < 0.05; Figure 8E), and there was no significant difference in the expressions between the pituitary gland and gonads (*p* > 0.05; Figure 8E). The expression of the *CYP17A2* gene in the pituitary gland of *S. curriculus* was significantly higher than that in the hypothalamus and gonads, and the expression of this gene in the gonads was significantly higher than that in the hypothalamus (*p* < 0.05; Figure 8E). There was no significant difference in the expression of the *CYP17A2* gene in the hypothalamus and pituitary gland of the hybrid F_1_ offspring, but both showed significantly higher expression of this gene than that in the gonads (*p* < 0.05; Figure 8E). There was no significant difference in the expressions of the *CYP17A2* gene in the hypothalamus of *C. idella* and the hybrid F_1_ offspring, whereas the expressions were significantly higher than that in *S. curriculus*. In the pituitary gland, the expression of the *CYP17A2* gene was the highest in *S. curriculus*, followed by that in the hybrid F_1_ offspring, and the lowest in the *C. idella*, and all of them had significant differences (*p* < 0.05; Figure 8E). In the gonads, the expression of the *CYP17A2* gene was the highest in *S. curriculus*, being significantly higher than that in *C. idella* and the hybrid F_1_ offspring, and there was no significant difference in the expressions of this gene in *C. idella* and hybrid F_1_ offspring (*p* > 0.05; Figure 8E).

The expression of the *CYP1B1* gene was the highest in the pituitary gland of *C. idella*, followed by that in the hypothalamus, and the lowest in the gonads, and all of them had significant differences (*p* < 0.05; Figure 8F). There was no significant difference in the expressions of the *CYP1B1* gene in the hypothalamus, pituitary, and gonads between *S. curriculus* and the hybrid F_1_ offspring (*p* > 0.05; Figure 8F). The expressions of the *CYP1B1* gene in the hypothalamus and gonads of *C. idella* were not significantly different from those of *S. curriculus* and the hybrid F_1_ offspring (*p* > 0.05; Figure 8F). The expression of the *CYP1B1* gene was the highest in the pituitary gland of *C. idella*, being significantly higher than that in *S. curriculus* and the hybrid F_1_ offspring (*p* < 0.05; Figure 8F), and there was no significant difference in the expression of this gene in *S. curriculus* and the hybrid F_1_ offspring in the pituitary gland (*p* > 0.05; Figure 8F).

The expression of the *CYP2AA12* gene in the hypothalamus of *C. idella* was significantly higher than that in the pituitary gland and gonads, and the expression in the pituitary was significantly higher than that in the gonads (*p* < 0.05; Figure 8G). The expression of the *CYP2AA12* gene in the hypothalamus of *S. curriculus* was significantly higher than that in the pituitary gland and gonads, and there was no significant difference in the expressions of this gene in the pituitary gland and gonads of *S. curriculus*. The expression of the *CYP2AA12* gene in the hypothalamus of the hybrid F_1_ offspring was not significantly different from that in the pituitary gland (*p* > 0.05; Figure 8G), whereas it was significantly higher than that in the gonads (*p* < 0.05; Figure 8G). The expression of the *CYP2AA12* gene in the hypothalamus of *S. curriculus* was significantly higher than that in *C. idella* (*p* < 0.05; Figure 8G), whereas there was no significant difference from that in the hybrid F_1_ offspring (*p* > 0.05; Figure 8G), and the expressions of this gene in the hypothalamus of *C. idella* and the hybrid F_1_ offspring were not significantly different (*p* > 0.05; Figure 8G). The expression of the *CYP2AA12* gene in the pituitary gland of *C. idella* was not significantly different from that in *S. curriculus* (*p* > 0.05; Figure 8G), whereas it was significantly lower than that in the hybrid F_1_ offspring (*p* < 0.05; Figure 8G). The expressions of the *CYP2AA12* gene in the gonads of *C. idella*, *S. curriculus*, and the hybrid F_1_ offspring were not significantly different (*p* > 0.05; Figure 8G).

The expression of the *UGT2A1* gene in the hypothalamus of *C. idella* was not significantly different to that in the gonads (*p* > 0.05; Figure 8H), whereas it was significantly lower than that in the pituitary gland (*p* < 0.05; Figure 8H). There was no significant difference in the expression of the *UGT2A1* gene in the hypothalamus, pituitary gland, and gonads of *S. curriculus* (*p* > 0.05; Figure 8H) or in the hybrid F_1_ offspring (*p* > 0.05; Figure 8H). There was no significant difference in the expression of the *UGT2A1* gene in the hypothalamus of *C. idella*, *S. curriculus*, or the hybrid F_1_ offspring (*p* > 0.05; Figure 8H). The expression of the *UGT2A1* gene in the pituitary gland of *S. curriculus* and the hybrid F_1_ offspring showed no significant differences (*p* > 0.05; Figure 8H), but it was significantly lower than that in *C. idella* (*p* < 0.05; Figure 8H). The expression of the *UGT2A1* gene in the gonads of the hybrid F_1_ offspring was not significantly different from that of *C. idella* or *S. curriculus* (*p* > 0.05; Figure 8H), whereas the expression of this gene in the gonads of *C. idella* was significantly higher than that in the gonads of *S. curriculus* (*p* < 0.05; Figure 8H).

There was no difference in the expression of the *UGT1A1* gene in the hypothalamus, pituitary gland, or gonads of *C. idella* (*p* > 0.05; Figure 8I). The expression of the *UGT1A1* gene in the hypothalamus of *S. curriculus* was not significantly different from that in the pituitary gland (*p* > 0.05; Figure 8I), while it was significantly lower than that in the gonads (*p* < 0.05; Figure 8I). The expression of the *UGT1A1* gene in the hypothalamus of the hybrid F_1_ offspring was not significantly different from that in the pituitary gland (*p* > 0.05; Figure 8I), whereas the expression in both was significantly lower than that in the gonads (*p* < 0.05; Figure 8I). There was no significant difference in the expression of the *UGT1A1* gene in the hypothalamus and pituitary gland between *C. idella*, *S. curriculus*, and the hybrid F_1_ offspring (*p* > 0.05; Figure 8I). The expression of the *UGT1A1* gene in the gonads of *C. idella* was significantly lower than that in the gonads of *S. curriculus* and the hybrid F_1_ offspring, and the expression of this gene in the gonads of *S. curriculus* was significantly lower than that in the hybrid F_1_ offspring’s gonads (*p* < 0.05; Figure 8I).

The expression of the *FSHR* gene was the highest in the pituitary gland of *C. idella*, followed by that in the gonads, and lowest in the hypothalamus, and all of the expression levels had significant differences (*p* < 0.05; Figure 8J). There was no significant difference in the expressions of the *FSHR* gene in the hypothalamus, pituitary, and gonad of *S. curriculus* (*p* > 0.05; Figure 8J). There was no significant difference in the expression of the *FSHR* gene in the hybrid F_1_ offspring’s pituitary gland and gonads (*p* > 0.05; Figure 8J), whereas it was significantly higher in the hypothalamus (*p* < 0.05; Figure 8J). There was no significant difference in the expressions of the *FSHR* gene in the hypothalamus of the three fishes (*p* > 0.05; Figure 8J). There was no significant difference in the expressions in the pituitary gland in *S. curriculus* and the hybrid F_1_ offspring (*p* > 0.05; Figure 8J), whereas they were significantly lower than the expression in the pituitary gland in *C. idella* (*p* < 0.05; Figure 8J). There was no significant difference in the expression of the *FSHR* gene in the gonads of *S. curriculus* and the hybrid F_1_ offspring (*p* > 0.05; Figure 8J), whereas it was significantly lower in the gonads of *C. idella* (*p* < 0.05; Figure 8J).

## 3. Discussion

Fish sex hormones are classified into three categories: protein hormones, glycoprotein hormones, and sex steroid hormones. Protein hormones include adrenocorticotropic hormone (ACTH), gonadotrophic growth hormone (GtH), thyroid-stimulating hormone (TSH), prolactin (PRL), growth hormone (GH), and melanocyte-stimulating hormone (MSH) [21]. Glycoprotein hormones mainly consist of follicle-stimulating hormone (FSH), luteinizing hormone (LH), GtH, and human chorionic gonadotropin (HCG) [22]. Sex steroids, such as 17β-estradiol (E2), 11-testosterone (11-KT), testosterone, estrone (E1), estriol (E3), progesterone, 17α-20β-dihydroxy-4-pregnen-3-one (DHP), and 17α-hydroxyprogesterone (17α-OHP), play a crucial role in inducing oocyte development and promoting maturation (meiosis) [23,24]. 11-KT is particularly important in spermatogenesis, sperm fertilization, and sperm storage [25,26]. The expression of serum 11-ketotestosterone reflects the degree of sperm nest development [27,28]. In fish, external stimulation triggers the hypothalamus to secrete GnRH, which then stimulates the pituitary gland to produce and release GtH (including FSH and LH). GtH travels through the bloodstream to the gonads, where it prompts the production of sex steroid hormones. These hormones, in turn, affect the development of eggs/sperm and regulate reproductive behavior [24,29]. GtH also regulates the secretion of sex hormones (mainly testosterone and E2), the initiation of the reproductive cycle, and the differentiation of germ cells. Sex hormones, in turn, regulate the synthesis and secretion of GnRH and GtH through negative feedback [30]. Fish have the ability to undergo sexual reversal by blocking the synthesis of exogenous hormones [31]. For instance, providing androgen to *Oncorhynchus mykiss* inhibits the synthesis of estrogen and results in virilization [32]. Sex steroids also play a crucial role in the process of sex reversal in species such as *Epinephelus akaara* [33], *Oreochromis mossambicus* [34], and *Oryzias latipes* [35]. Therefore, sex steroid hormones are essential for the development of gonadal differentiation. Our results suggested that insufficient transcription of genes involved in oocyte meiosis may be the main reason for the reduced reproductive ability of hybrid F_1_ offspring.

Transcriptome-sequencing technology has been widely applied to research on fish reproduction. In a study by Lin et al. [36], 12 key candidate genes related to sex determination and gonadal differentiation were identified through the transcriptome sequencing of mature gonadal tissues of *Symphysodon haraldi*. Similarly, He [37] used transcriptome sequencing to screen 19 genes related to the sex steroid hormone synthesis pathway and its receptor genes in stage III sperm nest and ovarian tissues of *Scatophagus argus*. The results showed that *CYP11A1*, *CYP11B2*, *CYP19A1B*, *HSD11B2*, *HSD3B1*, and *HSD3B7* genes were overexpressed in the sperm nest, whereas *CYP19A1A*, *HSD17B1*, *HSD17B8*, *HSD17B12*, and *HSD17B14* genes were overexpressed in the ovaries. In a study by Tao et al. [38], transcriptome sequencing was performed on male and female *Oreochromis niloticus* specimens that were exposed to high temperatures. The results showed that the expressions of genes related to androgen synthesis, such as *HSD17B7* and *3β-HSD*, increased in the male high-temperature treatment group, whereas the expression of estrogen synthesis genes, such as *CYP19A1A*, decreased. This suggests that the synthesis of sex hormones may play a role in the process of sexual reversal in *O. niloticus* under high-temperature treatment. In a study by Qin [39], transcriptome sequencing was performed on the brain and gonadal tissues of pseudomale, gynogenetic, and normal male and female *Nibea albiflora*. The results showed that male-related genes, such as *dmrt1*, *Gsdf*, *Amh*, and *Ar*, and female-related genes, such as *CYP19A*, *zp3*, *zp4*, and *foxl2*, were identified. Interestingly, *dmrt1* was only expressed in the sperm nest, whereas *CYP19A* was only expressed in the ovary.

The hypothalamic–pituitary–gonadal (HPG) axis is a crucial reproductive axis for studying sexual maturation and development. The neural and endocrine systems, with HPG as the core, primarily regulate gonadal development and gamete maturation in fish [40,41]. By conducting a differential expression analysis of the HPG axis in *C. idellus*, *S. curriculus*, and the hybrid F_1_ offspring, we accumulated enough data to analyze the molecular mechanisms underlying the difference in sexual maturity times between these two species and the reproductive disorders in the hybrid F_1_ offspring. Our study analyzed the HPG transcriptomes of these three fish species and identified several pathways associated with reproduction, including the oxytocin signaling pathway (ko04921), the GnRH signaling pathway, SNARE interactions in vesicular transport (ko04130), renin secretion (ko04924), and oocyte meiosis (ko04114). These results provide valuable omics information on *C. idellus*, *S. curriculus*, and their hybrid F_1_ offspring and serve as a reference for the further analysis of reproductive development in cyprinid fishes.

*HSD3B7*, *HSD17B1*, *HSD17B3*, and *HSD20B2* belong to the short-chain dehydrogenation/reductase (SDR) superfamily, which plays key roles in steroid hormones, biological metabolism, and redox sensing mechanisms. HSD3βs is involved in the oxidation and reduction of steroid hormones, and the expression pattern of HSD3βs is closely related to the growth-and-development period of animals. Among HSD3βs, *HSD3B7* plays a crucial role in the biosynthesis of all hormonal steroids. In the tilapia genome, two HSD3β genes have been identified, which may have an important impact on gonadal differentiation in tilapia [42]. HSD17βs affect the function of sex steroid hormones by regulating the binding of sex steroid hormones to receptors and controlling the expression of sex steroid hormones [43]. In Osteichthyes, *HSD17B1* catalyzes the transition between estrogen ketone and estrogen [44] and may also be involved in gonadal differentiation and development through sex steroid hormones [45].

*CYP11A1* is the first step in the synthesis of sex steroids and catalyzes the conversion of progesterone to pregnenolone. This gene has been cloned and identified in several fish species, including *Oryzias latipes* [46], *Odontesthes bonariensis* [47], *Danio rerio* [48], and *Anguilla japonica* [49], and is primarily expressed in the ovaries. It is believed to play a crucial role in oocyte development [50]. CYP17 is a microsomal cytochrome P450 enzyme that promotes the production of sex steroid hormones and cortisol. This gene has been cloned and identified in *Sebastods schlegelii*, *Paralichthys olivaceus*, *Verasper moseri,* and *Cynoglossus semilaevis* [10,51,52,53]. CYP19A1 has been identified as an early biomarker of ovarian differentiation in fish [50] and isolated in various bony fish species [54,55]. 3β-HSD is a gene that encodes 3β-hydroxysteroid dehydrogenase, which is involved in the conversion of sterol hormones in hormone-producing tissues. This gene has been cloned in the genomes of *Danio rerio*, *Oreochromis mossambicus*, and *Oryzias latipes* [56,57]. In bony fishes, Ad4BP/sf1 binds to the CYP19A promoter and affects the expression characteristics of aromatase genes, thereby regulating the synthesis of sex hormones [58]. sf1 binding sites in the promoter region of CYP19A1A have been found in *Gobiocypris rarus*, *Oreochromis niloticus*, and *Lateolabrax japonicus* [59,60]. In tilapia, the expression of CYP19A1A in the gonads of females decreases with the inhibition of *foxl2* expression, resulting in a decrease in serum E2 levels and ultimately leading to the induction of male characteristics [61]. Mutations in foxl2 or CYP19A1A can cause sexual reversal from females to males [62]. In zebrafish, double mutants of the two subtypes of foxl2a and foxl2b can result in complete female sex reversal in the early stages [63].

In this study, we observed significant differences in the expression of *HSD3B7*, *HSD17B1*, *HSD17B3*, *HSD20B2*, *CYP17A2*, *CYP1B1*, *CYP2AA12*, *UGT2A1*, *UGT1A1*, and *FSHR* in the hypothalamus, pituitary, and gonadal tissues of *C. idella*, *S. curriculus*, and their hybrid F_1_ offspring. Specifically, the expression of *UGT1A1* was significantly higher in the gonads of the hybrid F_1_ offspring compared to that in *C. idella* and *S. curriculus*. Additionally, the expressions of *HSD3B7* and *CYP17A2* in the pituitary gland and gonads of *S. curriculus* were significantly higher than those in *C. idella* and the hybrid F_1_ offspring. Furthermore, the expression of *CYP2AA12* in the hypothalamus of *S. curriculus* was significantly higher than that in *C. idella* and the hybrid F_1_ offspring, with the most significant expression observed in the pituitary gland of the hybrid F_1_ offspring. With the exception of *UGT1A1*, the expressions of the remaining nine genes in the pituitary tissues of the three species were significantly different. Specifically, the expressions of *HSD17B1*, *HSD17B3*, *HSD20B2*, *CYP1B1*, *UGT2A1*, and *FSHR* in the pituitary tissues of *C. idella* were significantly higher than in *S. curriculus* and the hybrid F_1_ offspring. Further research is needed to determine which of these genes are the key factors in regulating F_1_ gonadal development in the hybrid *C. idella* (♀) × *S. curriculus* (♂) and the related molecular mechanisms that regulate the synthesis of steroid hormones and affect the differentiation and development of gonadal differentiation.

## 4. Materials and Methods

### 4.1. Experimental Design and Sample Collection

The animal experiments were conducted in accordance with the guidelines approved by the Animal Care and Use Committee of Hunan Agricultural University (Changsha, China; Approval Code: 201903295; Approval Date: 13 September 2019). Based on the results of our previous histological study on hybrid F_1_ gonadal development [20], 150-day-old specimens of *C. idella* (Gc), *S. curriculus* (Sc), and their hybrid F_1_ offspring (Zj) were collected from Xiangyin Institute of Fishery Sciences in Hunan Province in October 2019. The fish were cultured in different cages in an indoor circulating water culture system with an average water temperature of 26.0 °C and containing 6.5 mg/L of dissolved oxygen. The fish were fed 3% of their average body weight in the same commercial feed twice daily (8:00 and 18:00). Fifteen larvae from each species were randomly collected from the cages and anesthetized with 200 mg/L of tricaine methanesulfonate (MS-222). Due to the small size of the fish and the small quantity of tissue samples, five fish tissues of each species were mixed to create a sample for transcriptome sequencing. Three samples were collected for each species, with separate samples for the hypothalamus (xqn), pituitary (ct), and gonadal (xx) tissues. The samples were quickly frozen using liquid nitrogen and stored at −80 °C (Appendix A).

### 4.2. Total RNA Extraction and Transcriptome Sequencing

Total RNA was extracted from fish tissues using a TRIzol RNA extraction kit (Omega Bio-tek, Norcross, GA, USA). To enrich for mRNA, magnetic beads containing Oligo^dT^ (Yeasen Biotechnology, Shanghai, China) were used, followed by random fragmentation using reagents (Thermo Fisher Scientific, Waltham, MA, USA). The resulting eukaryotic mRNA was then used as a template for cDNAs synthesis and purified using AMPure XP beads (Beckman Coulter, Brea, CA, USA). Transcriptome sequencing was performed using the Illumina HiSeq platform at Biomarker Technologies Co., Ltd. (Beijing, China), following the previously described method [64].

After checking for redundancy, the integrity of deredundant transcriptome was evaluated using BUSCO [65]. Single-copy gene sets from multiple evolutionary branches were then constructed using the BUSCO-referenced OrthoDB database to assess the accuracy and completeness of the transcripts. The coding sequences (CDSs) were analyzed using TransDecoder. The LncRNAs of the transcriptomes were analyzed and predicted using CPC [66], CNCI, pfam protein domain, and CPAT. The transcript sequences were then compared to the animalTFDB 2.0 database [67] to identify any potential transcription factors. Non-redundant transcript sequences were also compared to the NR [68], Swissprot [69], GO [70], COG [71], KOG, Pfam [72], and KEGG [73] databases using BLAST [74] to obtain functional annotations for the transcripts. Transcript expression analysis was performed using RSEM [75], and differential expression analysis was conducted using DESeq [76].

### 4.3. RT-qPCR

The expression levels of 10 gonadal developmental genes (*HSD3B7*, *HSD17B1*, *HSD17B3*, *HSD20B2*, *CYP17A2*, *CYP1B1*, *CYP2AA12*, *UGT2A1*, *UGT1A1*, and *FSHR*; Table 2) in the hypothalamus, pituitary, and gonadal tissues were detected using real-time qPCR as previously described [77].

### 4.4. Data Analysis

Data were presented as means ± standard deviation (SD). To analyze the data, a one-way ANOVA was conducted using R 4.2.3 [78]. Statistical significance was set at a *p*-value of less than 0.05.

## 5. Conclusions

After conducting transcriptome analysis on *C. idella*, *S. curriculus*, and their hybrid F_1_ offspring, several pathways related to reproduction were identified. These mainly included the oxytocin signaling pathway (ko04921), the GnRH signaling pathway, SNARE interactions in vesicular transport (ko04130), renin secretion (ko04924), and oocyte meiosis (ko04114). The insufficient transcription of genes involved in oocyte meiosis was found to be the main factor contributing to the inadequate reproductive ability of the hybrid F_1_ offspring. Through transcriptome analysis, a total of 10 key genes responsible for gonadal development were identified, including *HSD3B7*, *HSD17B1*, *HSD17B3*, *HSD20B2*, *CYP17A2*, *CYP1B1*, *CYP2AA12*, *UGT2A1*, *UGT1A1*, and *FSHR*. These genes showed varying expression patterns in different tissues of *C. idella*, *S. curriculus*, and their hybrid F_1_ offspring.

## Figures and Tables

**Figure 1 ijms-25-10566-f001:**
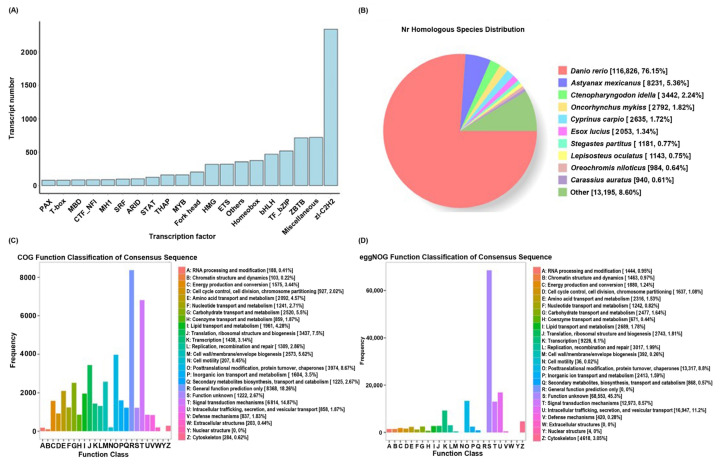
Basic transcriptome results. (**A**) Distribution of transcription factor types. The horizontal axis represents the predicted transcription factor type, and the vertical axis represents the predicted number. (**B**) Species classification of transcripts annotated with NR. (**C**) Statistical chart regarding COG annotation classification of transcripts. The horizontal axis represents the classification content of COG, and the vertical axis represents the number of transcripts. (**D**) Statistical chart regarding eggnog annotation classification of transcripts.

**Figure 2 ijms-25-10566-f002:**
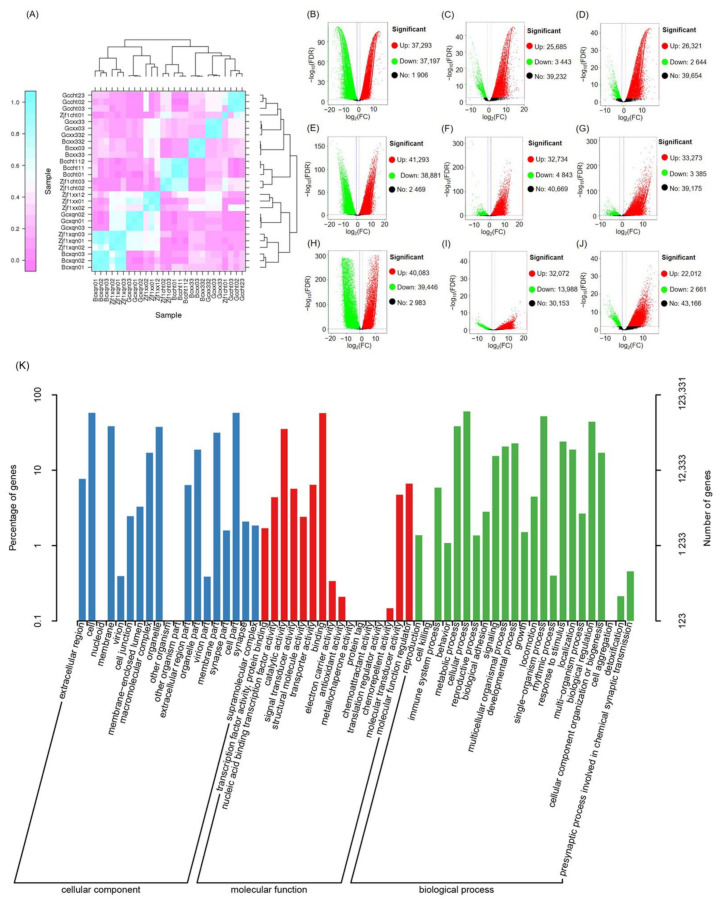
Heatmap profile (**A**) and volcano plots (**B**–**J**) displaying the variations in gene transcriptional levels between *C. idella*, *S. curriculus*, and their hybrid F_1_ offspring; (**B**) number of genes with significant differences in hypothalamus transcription between *C. idella* and *S. curriculus*; (**C**) number of genes with significant differences in hypothalamus transcription between their hybrid F_1_ offspring and *S. curriculus*; (**D**) number of genes with significant differences in hypothalamus transcription between their hybrid F_1_ offspring and *C. idella*; (**E**) number of genes with significant differences in pituitary transcription between *C. idella* and *S. curriculus*; (**F**) number of genes with significant differences in pituitary transcription between their hybrid F_1_ offspring and *S. curriculus*; (**G**) number of genes with significant differences in pituitary transcription between their hybrid F_1_ offspring and *C. idella*; (**H**) number of genes with significant differences in gonadal transcription between *C. idella* and *S. curriculus*; (**I**) number of genes with significant differences in gonadal transcription between their hybrid F_1_ offspring and *S. curriculus*; (**J**) number of genes with significant differences in gonadal transcription between their hybrid F_1_ offspring and *C. idella*; (**K**) GO classification of differential genes.

**Figure 3 ijms-25-10566-f003:**
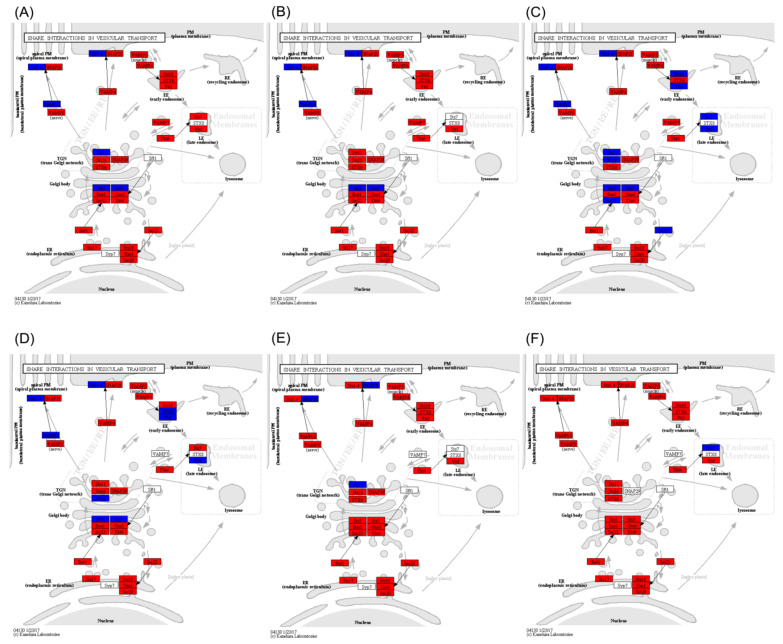
Differences in gonadal-development-related pathways in hypothalamus (**A**,**D**), pituitary (**B**,**E**), and gonadal (**C**,**F**) tissues between *S. curriculus* and hybrid F_1_ offspring (**A**–**C**) and between *C. idella* and hybrid F_1_ offspring (**D**–**F**). Red indicates pathways that contain significantly up-regulated genes in hybrid F_1_ offspring compared with *S. curriculus* or *C. idella*, green indicates pathways that contain significantly down-regulated genes in hybrid F_1_ offspring compared with *S. curriculus* or *C. idella*, and blue indicates pathways that contain significantly down- and up-regulated genes in hybrid F_1_ offspring compared with *S. curriculus* or *C. idella*.

**Figure 4 ijms-25-10566-f004:**
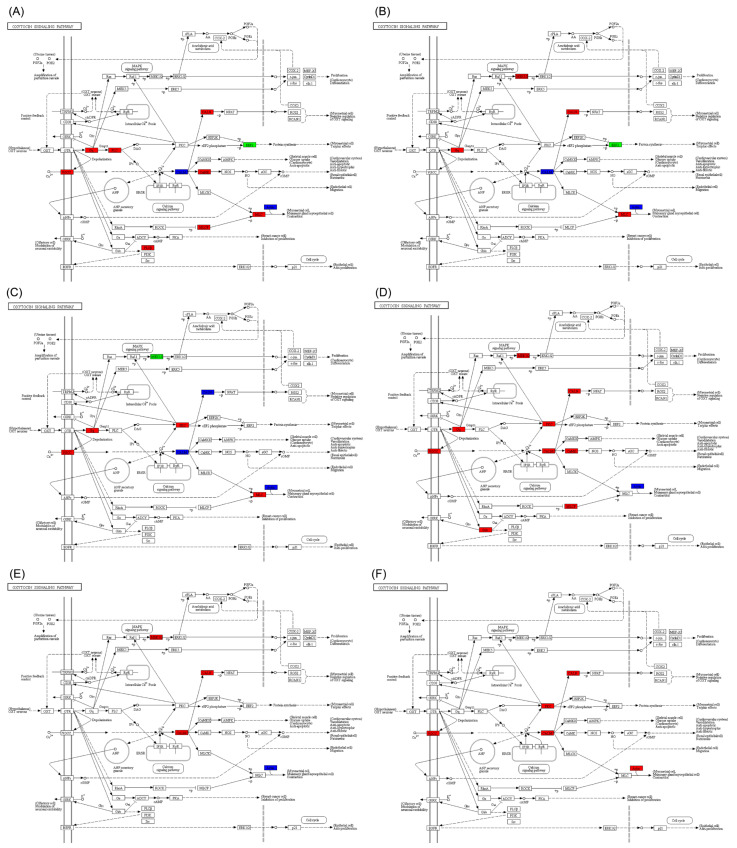
Differences in oxytocin signaling pathway in hypothalamus (**A**,**D**), pituitary (**B**,**E**), and gonadal (**C**,**F**) tissues between S. curriculus and hybrid F_1_ offspring (**A**–**C**) and between *C. idella* and hybrid F_1_ offspring (**D**–**F**). Red indicates pathways that contain significantly up-regulated genes in hybrid F_1_ offspring compared with *S. curriculus* or *C. idella*, green indicates pathways that contain significantly down-regulated genes in hybrid F_1_ offspring compared with *S. curriculus* or *C. idella*, and blue indicates pathways that contain significantly down- and up-regulated genes in hybrid F_1_ offspring compared with *S. curriculus* or *C. idella*.

**Figure 5 ijms-25-10566-f005:**
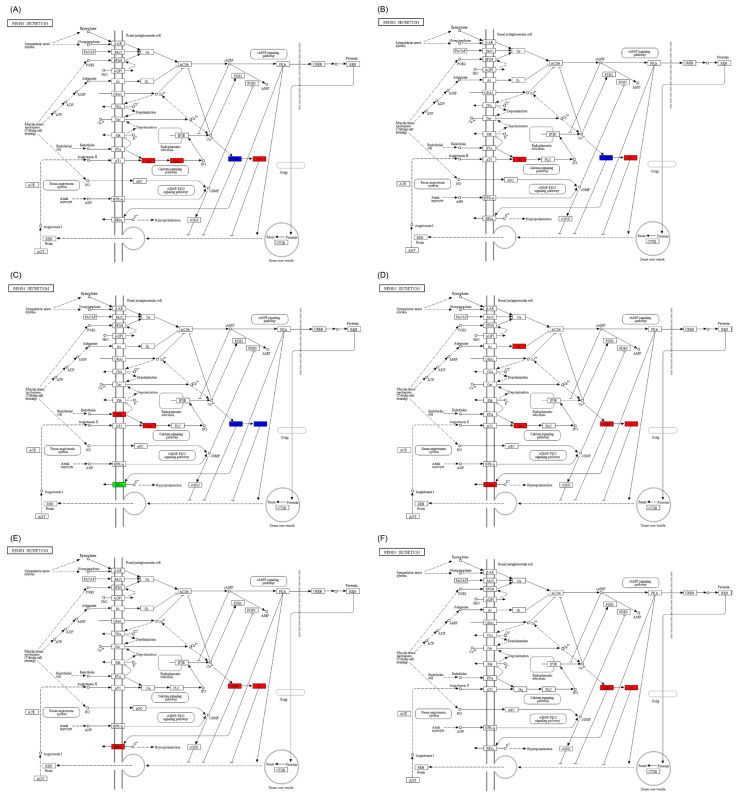
Differences in renin secretion pathway in hypothalamus (**A**,**D**), pituitary (**B**,**E**), and gonadal (**C**,**F**) tissues between *S. curriculus* and hybrid F_1_ offspring (**A**–**C**) and between *C. idella* and hybrid F_1_ offspring (**D**–**F**). Red indicates pathways that contain significantly up-regulated genes in hybrid F_1_ offspring compared with *S. curriculus* or *C. idella*, green indicates pathways that contain significantly down-regulated genes in hybrid F_1_ offspring compared with *S. curriculus* or *C. idella*, and blue indicates pathways that contain significantly down- and up-regulated genes in hybrid F_1_ offspring compared with *S. curriculus* or *C. idella*.

**Figure 6 ijms-25-10566-f006:**
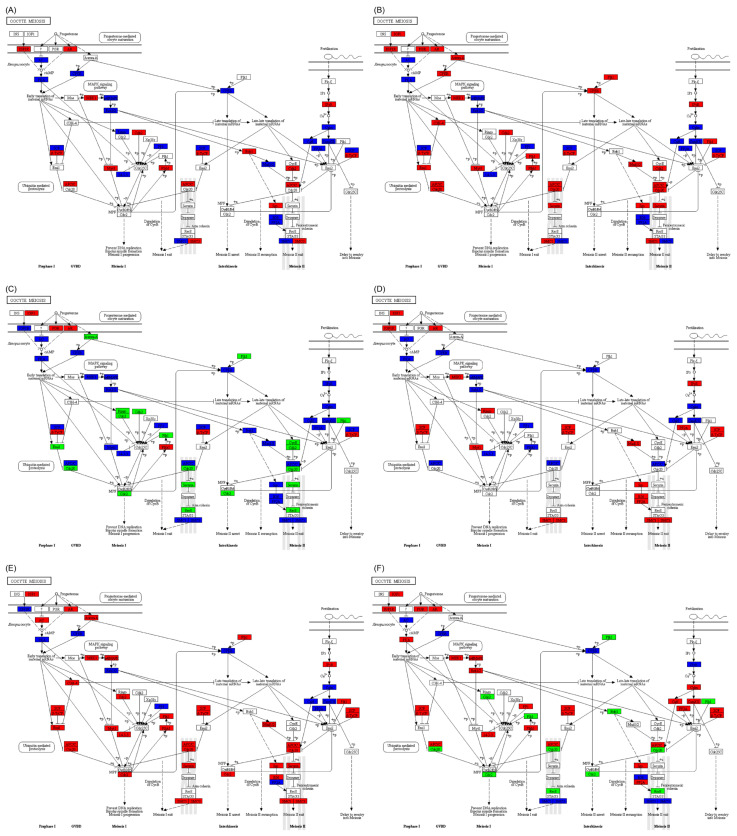
Differences in oocyte meiosis pathway in hypothalamus (**A**,**D**), pituitary (**B**,**E**), and gonadal (**C**,**F**) tissues between S. curriculus and hybrid F_1_ offspring (**A**–**C**) and between C. idella and hybrid F_1_ offspring (**D**–**F**). Red indicates pathways that contain significantly up-regulated genes in hybrid F_1_ offspring compared with *S. curriculus* or *C. idella*, green indicates pathways that contain significantly down-regulated genes in hybrid F_1_ offspring compared with *S. curriculus* or *C. idella*, and blue indicates pathways that contain significantly down- and up-regulated genes in hybrid F_1_ offspring compared with S. curriculus or *C. idella*.

**Figure 7 ijms-25-10566-f007:**
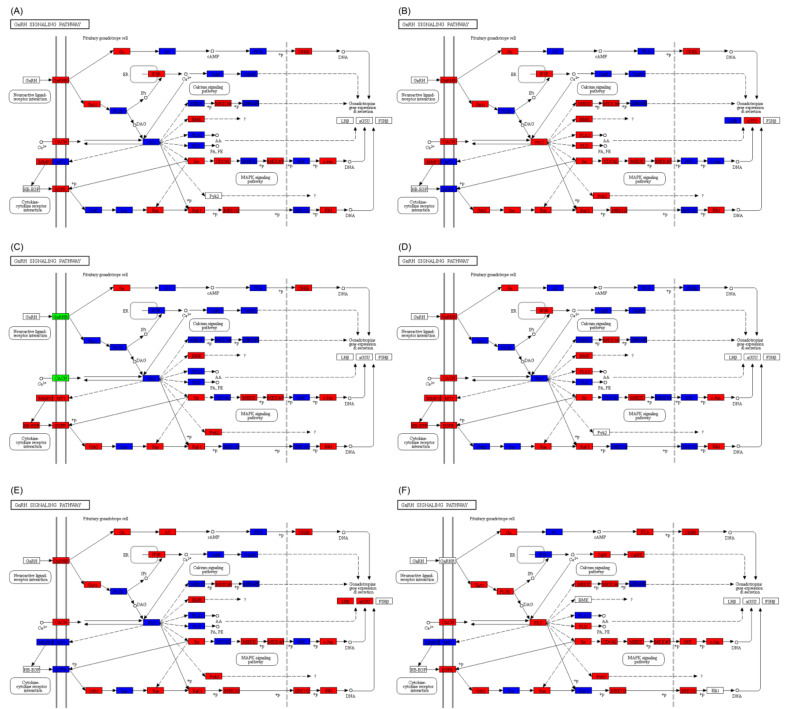
Differences in GnRH signaling pathway in hypothalamus (**A**,**D**), pituitary (**B**,**E**), and gonadal (**C**,**F**) tissues between *S. curriculus* and hybrid F_1_ offspring (**A**–**C**) and between *C. idella* and hybrid F_1_ offspring (**D**–**F**). Red indicates pathways that contain significantly up-regulated genes in hybrid F_1_ offspring compared with *S. curriculus* or *C. idella*, green indicates pathways that contain significantly down-regulated genes in hybrid F_1_ offspring compared with *S. curriculus* or *C. idella*, and blue indicates pathways that contain significantly down- and up-regulated genes in hybrid F_1_ offspring compared with *S. curriculus* or *C. idella*.

**Figure 8 ijms-25-10566-f008:**
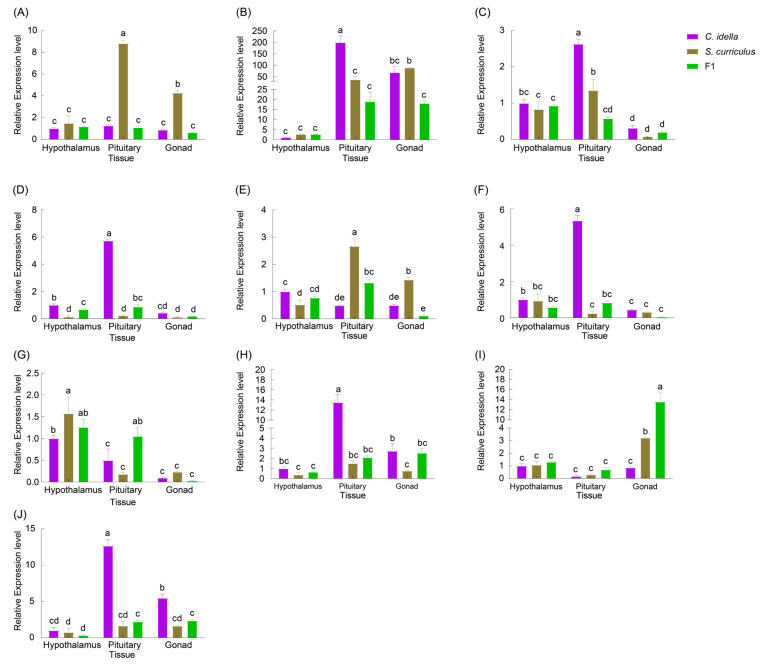
Expression of gonadal-development-related genes in *C. idella*, *S. curriculus*, and the hybrid F_1_ offspring. (**A**) Relative expression level of *HSD3B7*; (**B**) relative expression level of *HSD17B1*; (**C**) relative expression level of *HSD17B3*; (**D**) relative expression level of *HSD20B2*; (**E**) relative expression level of *CYP17A2*; (**F**) relative expression level of *CYP1B1*; (**G**) relative expression level of *CYP2AA12*; (**H**) relative expression level of *UGT2A1*; (**I**) relative expression level of *UGT1A1*; (**J**) relative expression level of *FSHR*. Different lowercase letters above the bars indicate there are significant differences between the groups.

**Table 1 ijms-25-10566-t001:** Gene enrichment of gonadal-development-related pathways.

Fish	Pathway Term	ko ID	Rich Factor	q-Value	Gene Number
Scct_vs_Zjf1ct	SNARE interactions in vesicular transport	ko04130	1.417349	0.445305	64
	Oxytocin signaling pathway	ko04921	1.499386	1	34
	Renin secretion	ko04924	1.635953	1	20
Scxqn_vs_Zjf1xqn	Oocyte meiosis	ko04114	1.210491	0.001366	413
	GnRH signaling pathway	ko04912	1.221223	0.001727	371
	SNARE interactions in vesicular transport	ko04130	1.349169	0.607791	76
	Oxytocin signaling pathway	ko04921	1.414003	1	40
	Renin secretion	ko04924	1.573648	1	24
Gcct_vs_Scct	SNARE interactions in vesicular transport	ko04130	1.343209	6.61 × 10^−5^	150
Gcxqn_vs_Scxqn	Oocyte meiosis	ko04114	1.179919	2.62 × 10^−10^	846
	GnRH signaling pathway	ko04912	1.121513	0.001189	716
	SNARE interactions in vesicular transport	ko04130	1.241767	0.025536	147
Gcxqn_vs_Zjf1xqn	Oocyte meiosis	ko04114	1.23236	0.000173	416
	GnRH signaling pathway	ko04912	1.221012	0.001951	367
Gcxx_vs_Scxx	SNARE interactions in vesicular transport	ko04130	1.284499	0.0018	150

**Table 2 ijms-25-10566-t002:** The RT-qPCR primers used in this study.

Target Gene	Primer Name	Primer Sequence 5′-3′
*HSD3B7*	HSD3B7-F	ACAAAGTGTGGCAACTTGGC
	HSD3B7-R	TCACACCAATAGGCTGCTTG
*HSD17B1*	HSD17B1-F	TGGACCAGTCAACACAGACTTC
	HSD17B1-R	TGAGCTGCATTCTGGAACAC
*HSD17B3*	HSD17B3-F	ATTCTGCCCAGCCAAATACC
	HSD17B3-R	TTTGCTGCATTCCTGGTAGC
*HSD20B2*	HSD20B2-F	GCGACAGACACATGTGATTCAG
	HSD20B2-R	TCCATGCCCATTAGCTGTTG
*CYP17A2*	CYP17A2-F	ACGCCGTTCTTTGTGAAGTG
	CYP17A2-R	TTGTGTCCTGCATAGCAACG
*CYP1B1*	CYP1B1-F	TCGCTTCATTTCGGTTCGTG
	CYP1B1-R	TGTTTGGTGTGGATGTTGGC
*CYP2AA12*	CYP2AA12-F	ACCCAGATGTACAAGAGCGATG
	CYP2AA12-R	TTGCCAAAGCGCTGAAACTC
*UGT2A1*	UGT2A1-F	TGCCTTACACAAAGCAGGAC
	UGT2A1-R	TGGAAGCCGTGATGATGTTG
*UGT1A1*	UGT1A1-F	TTCCCCAAACCTCAAATGCC
	UGT1A1-R	TGAAGACCACAAAGCCATGC
*FSHR*	FSHR-F	TTCTCACGCCAAAGTCTTGC
	FSHR-R	TGTTTTGAAGCAGCCGAACC

## Data Availability

The data that support the findings of this study are available in the Sequence Read Archive (SRA) under the accession number PRJNA1146714.

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
