# Peer review of "Gonadal Development and Differentiation of Hybrid F_1_ Line of *Ctenopharyngodon idella* (♀) × *Squaliobarbus curriculus* (♂)"

_ijms, 2024, doi:10.3390/ijms251910566_

Round 1

Reviewer 1 Report

Comments and Suggestions for Authors

I have read the manuscript carefully; I found that manuscript was not well organized, the language is poor, making it unreadable. The main problems were as the following:

1. Abstract should display the persuasive results directly, i.e. which genes/signals and how cause the gonadal development retardation of hybrid F1. It difficult to get meaningful information form the abstract. In addition, the key words selection also improper.

2. Introduction have provided some background and include all relevant references, but it difficult to get the research purpose.

3. Some unrelated results were displayed in the results, for example, SSR analysis results. The author should analyze the results comprehensively, but not simple repetition listing, such as the content of page12.

Overall, I think the author should reorganize the article to make it concise and key results more salient.

Comments on the Quality of English Language

The language was not well organized, making the manuscript unreadable.

Author Response

Comments and Suggestions for Authors

I have read the manuscript carefully; I found that manuscript was not well organized, the language is poor, making it unreadable. The main problems were as the following:

  1. Abstract should display the persuasive results directly, i.e. which genes/signals and how cause the gonadal development retardation of hybrid F1. It difficult to get meaningful information form the abstract. In addition, the key words selection also improper.

Response

Thank you very much for your comments. These comments are vary important for us to revise our manuscript and improve its readability. We have revised the manuscript according to your comments. Thank you again. Although we totally agree with our comment that “abstract should display the persuasive results directly”, because too many genes were involved, and they did not show a completely consistent trend of change, it is difficult to describe the changes of these genes completely and directly in the Abstract section.

Comments and Suggestions for Authors

  1. Introduction have provided some background and include all relevant references, but it difficult to get the research purpose.

Response

Thank you for your comment. As described in the Introduction section of our manuscript, the purpose of our study was to uncover the molecular regulatory mechanism of gonadal development in hybrid F1 of C. idella (♀) × S. curriculus (♂).

Comments and Suggestions for Authors

  1. Some unrelated results were displayed in the results, for example, SSR analysis results. The author should analyze the results comprehensively, but not simple repetition listing, such as the content of page12.

Response

Thank you for your comment. We have deleted the SSR analysis according to your comment. Because of the different analytical techniques, although RT-qPCR is a further verification of the results of transcriptome sequencing, the results of these RT-qPCR were not completely consistent with the results of transcriptome sequencing because of the different analytical methods to judge whether there are significant differences. Therefore, in order to avoid misunderstanding, we think it is easier to understand the results obtained by the analysis method separately, and it is also easier for other researchers to compare our results horizontally according to the research method.

Comments and Suggestions for Authors

Overall, I think the author should reorganize the article to make it concise and key results more salient.

Response

Thank you for your comment. We have reorganized our manuscript according to your comments.

Comments on the Quality of English Language

The language was not well organized, making the manuscript unreadable.

Response

We have carefully checked the expression and language of our revised manuscript according to your comments. Thank you for your comment.

Reviewer 2 Report

Comments and Suggestions for Authors

This manuscript  „Gonadal Development and Differentiation of Hybrid F1 of Ctenopharyngodon idella () × Squaliobarbus curriculus ()” investigates the gonadal development and differentiation of hybrid F1 offspring of Ctenopharyngodon idella () × Squaliobarbus curriculus (). The authors found that the hybrid F1 offspring exhibit heterosis in disease resistance but also show abnormal sex differentiation. To understand the mechanism behind gonadal differentiation in the hybrid F1, the authors analyzed the transcriptomes of C. idella, S. curriculus, and the hybrid F1.

The study is well-designed and conducted. The authors present a comprehensive analysis of gene expression changes in the hypothalamus, pituitary, and gonads of the three fish species. They identify key molecules involved in gonad development and show significant differences in expression levels between the three species. The authors also provide evidence for the role of insufficient transcription of genes involved in oocyte meiosis in the reduced reproductive ability of hybrid F1 offspring.

This study is a valuable contribution to the field of fish reproductive biology and will be of interest to researchers studying the mechanisms of gonadal development and sex differentiation. However, I recommend that the authors make some revisions before publication.

Note on the title – the use of symbols () and () seems justified, but there are concerns that using such symbols in the title may cause a problem with searching for the article in publication databases or on the internet. The editor should decide whether using these symbols in the title would be beneficial.

The methods chosen are appropriate, the authors employed RNA sequencing and RT-qPCR]. However, information regarding the number of individuals studied and their rearing conditions is missing.

Line 85 – it is „softeare” – it should be „software”

Author Response

Comments and Suggestions for Authors

This manuscript  ”Gonadal Development and Differentiation of Hybrid F1 of Ctenopharyngodon idella (♀) × Squaliobarbus curriculus (♂)” investigates the gonadal development and differentiation of hybrid F1 offspring of Ctenopharyngodon idella (♀) × Squaliobarbus curriculus (♂). The authors found that the hybrid F1 offspring exhibit heterosis in disease resistance but also show abnormal sex differentiation. To understand the mechanism behind gonadal differentiation in the hybrid F1, the authors analyzed the transcriptomes of C. idella, S. curriculus, and the hybrid F1.

The study is well-designed and conducted. The authors present a comprehensive analysis of gene expression changes in the hypothalamus, pituitary, and gonads of the three fish species. They identify key molecules involved in gonad development and show significant differences in expression levels between the three species. The authors also provide evidence for the role of insufficient transcription of genes involved in oocyte meiosis in the reduced reproductive ability of hybrid F1 offspring.

This study is a valuable contribution to the field of fish reproductive biology and will be of interest to researchers studying the mechanisms of gonadal development and sex differentiation. However, I recommend that the authors make some revisions before publication.

Response

Thank you very much for your positive comments. We have revised our manuscript according to your comments, which have improved the readability of our manuscript and the repeatability of the experiments, and we are very grateful for this.

Comments and Suggestions for Authors

Note on the title – the use of symbols (♀) and (♂) seems justified, but there are concerns that using such symbols in the title may cause a problem with searching for the article in publication databases or on the internet. The editor should decide whether using these symbols in the title would be beneficial.

Response

Considering that the symbols (♀ and ♂) are included in the titles of many papers, such as Wang et al. 2018. Comparative transcriptome analysis between interspecific hybridization (Huaren apricot ♀ × almond ♂) and intraspecific hybridization (Huaren apricot) during young fruit developmental stage. Scientia Horticulturae, 240: 397-404; He et al. 2013. Insights into food preference in hybrid F1 of Siniperca chuatsi (♀) × Siniperca scherzeri (♂) mandarin fish through transcriptome analysis. BMC Genomics, 14: 601; Qin et al. 2016. Rapid genomic changes in allopolyploids of Carassius auratus red var. (♀) × Megalobrama amblycephala (♂). Scientific Reports, 6: 34417; Qin et al. 2018. Induced gynogenesis in autotetraploids derived from Carassius auratus red var. (♀) × Megalobrama amblycephala (♂). Aquaculture, 495: 710-714, we think it is acceptable to include the symbols in the title. And thus the source parent information of the hybrid offspring can be displayed more clearly.  

Comments and Suggestions for Authors

The methods chosen are appropriate, the authors employed RNA sequencing and RT-qPCR]. However, information regarding the number of individuals studied and their rearing conditions is missing.

Response

Thank you very much for your comment. As we described in the Materials and Methods, “due to the small size of the fish and the small amount of tissue samples, five fish tissues of each species were mixed to create a sample for transcriptome sequencing. Three samples were collected for each species, with separate samples for the hypothalamus (xqn), pituitary (ct), and gonadal (xx) tissues. The samples were quickly frozen using liquid nitrogen and stored at -80°C.” Furthermore, We have added the rearing conditions according to your comments.

Comments and Suggestions for Authors

Line 85 – it is “softeare” – it should be “software”

Response

Thank you for your comment. We have revised the spelling mistake according to your comment.

Round 2

Reviewer 1 Report

Comments and Suggestions for Authors

The manuscript has been revised accordingly, thus  can be accepted in the present form.